# Specific Cell Targeting by *Toxoplasma gondii* Displaying Functional Single-Chain Variable Fragment as a Novel Strategy; A Proof of Principle

**DOI:** 10.3390/cells13110975

**Published:** 2024-06-04

**Authors:** Muna Aljieli, Clément Rivière, Louis Lantier, Nathalie Moiré, Zineb Lakhrif, Anne-France Boussemart, Thomas Cnudde, Laurie Lajoie, Nicolas Aubrey, Elhadi M. Ahmed, Isabelle Dimier-Poisson, Anne Di-Tommaso, Marie-Noëlle Mévélec

**Affiliations:** 1BioMAP, UMR ISP 1282 INRAE, Université de Tours, 37200 Tours, France; muna.aljieli@etu.univ-tours.fr (M.A.); clement.riviere@univ-tours.fr (C.R.); louis.lantier@univ-tours.fr (L.L.); nathalie.moire@univ-tours.fr (N.M.); zineb.lakhrif@univ-tours.fr (Z.L.); anne-france.prouvost@orange.fr (A.-F.B.); laurie.lajoie@univ-tours.fr (L.L.); nicolas.aubrey@univ-tours.fr (N.A.); dimier@univ-tours.fr (I.D.-P.); mevelec@univ-tours.fr (M.-N.M.); 2Faculty of Pharmacy, University of Gezira, Wad Madani 21111, Sudan

**Keywords:** *Toxoplasma gondii*, surface display, targeting, scFv, cancer immunotherapy, immune checkpoint, PD-L1

## Abstract

*Toxoplasma gondii* holds significant therapeutic potential; however, its nonspecific invasiveness results in off-target effects. The purpose of this study is to evaluate whether *T. gondii* specificity can be improved by surface display of scFv directed against dendritic cells’ endocytic receptor, DEC205, and immune checkpoint PD-L1. Anti-DEC205 scFv was anchored to the *T. gondii* surface either directly via glycosylphosphatidylinositol (GPI) or by fusion with the SAG1 protein. Both constructs were successfully expressed, but the binding results suggested that the anti-DEC-SAG1 scFv had more reliable functionality towards recombinant DEC protein and DEC205-expressing MutuDC cells. Two anti-PD-L1 scFv constructs were developed that differed in the localization of the HA tag. Both constructs were adequately expressed, but the localization of the HA tag determined the functionality by binding to PD-L1 protein. Co-incubation of *T. gondii* displaying anti-PD-L1 scFv with tumor cells expressing/displaying different levels of PD-L1 showed strong binding depending on the level of available biomarker. Neutralization assays confirmed that binding was due to the specific interaction between anti-PD-L1 scFv and its ligand. A mixed-cell assay showed that *T. gondii* expressing anti-PD-L1 scFv predominately targets the PD-L1-positive cells, with negligible off-target binding. The recombinant RH-PD-L1-C strain showed increased killing ability on PD-L1+ tumor cell lines compared to the parental strain. Moreover, a co-culture assay of target tumor cells and effector CD8+ T cells showed that our model could inhibit PD1/PD-L1 interaction and potentiate T-cell immune response. These findings highlight surface display of antibody fragments as a promising strategy of targeting replicative *T. gondii* strains while minimizing nonspecific binding.

## 1. Introduction

*Toxoplasma gondii* is an obligate intracellular apicomplexan protozoan with a natural ability to infect all nucleated mammalian cells [1]. In humans, primary infection with *T. gondii* in immunocompetent individuals is predominantly asymptomatic. However, it becomes a significant concern in cases of materno-fetal transmission or in severely immunocompromised individuals [2]. It has been reported that *T. gondii* infection can stimulate a cascade of potent immunological responses characterized by the generation of specific CD4+ and CD8+ T cells that produce interferon gamma (IFN-γ) [3]. The potential therapeutic benefits of *Toxoplasma gondii* infection in various diseases, including cancer, have been reported for decades. Chronic infection and even parasite extracts have been shown to confer resistance in mice to numerous pathogens, including viruses [4], bacteria [5], parasites [6], yeast [7] and tumors [8,9].

It is noteworthy that, in the context of tumors, immunotherapy based on live microorganisms such as viruses and bacteria has emerged as a promising treatment strategy. Several microorganisms are currently undergoing preclinical and clinical trials to investigate their oncolytic potential [10,11]. Moreover, advances in genetic engineering have allowed the application of several strategies to enhance oncolysis and immune stimulation or to counteract the tumor neovascularization [12,13]. Additionally, several attenuated viruses and bacteria strains have been generated and widely investigated as vaccine vectors to elicit tumor antigen-specific CD8+ T-cell responses and induce protective immunity against tumor development [14,15]. In 2015, the U.S Food and Drug Administration (FDA) approved the use of the recombinant attenuated herpes simplex virus encoding human granulocyte macrophage colony-stimulating factor (GM-CSF) for treatment of melanoma patients with injectable but non-resectable lesions in the skin and lymph nodes [16].

In the context of microorganism-based immunotherapy, *T. gondii* emerges as a promising candidate for immunotherapeutic applications. However, to ensure its safe implementation, significant challenges must be addressed. Indeed, uncontrolled parasite reproduction, inflammatory processes, and damage to healthy tissues might all contribute to undesirable adverse effects. To address this challenge, attenuated strains have demonstrated considerable utility, facilitated by advancements in genetic manipulation tools developed since 1993 [17]. The first recombinant attenuated strain was a live, non-replicating parasite, named CPS (in which the Carbamoyl Phosphate Synthetase gene is disrupted) [18]. This strain successfully achieved tumor regressions in murine models of melanoma [19], ovarian cancer [20], and pancreatic cancer [21] following intratumoral injections of the parasite. The live parasites activate the immune system, counteracting immunosuppression and thereby reactivating the tumor’s specific adaptive immunity. Recently, more attenuated strains have been generated including those lacking lactate dehydrogenases, dense granule 17 (GRA17), or dense granule 5 (GRA5). These strains demonstrated effective anti-tumor effects in mice models [22,23,24].

Despite the successful application of attenuated strains, nonetheless, the invasiveness of the live *T. gondii* remains crucial for its anti-tumor activity [25]. Therefore, developing novel strategies allowing the use of replicative strains is also essential to enhance oncolysis and overcome the limitations of attenuation. One of these strategies is targeting tumor-associated antigens that are uniquely expressed or significantly overexpressed in tumor cells. This approach has been widely applied to several viruses [26] and bacteria [27], notably through expression of antibody fragments directed against tumor-specific surface antigens. Targeting strategies have emerged as promising avenues for future vaccines and therapies. In the context of vaccination, dendritic cells (DCs) represent one of the most optimal targets due to their potent ability to prime immune responses [28]. Among the various receptors suitable for targeting on DCs is the DEC205 receptor, which has been extensively studied for antigen delivery. We previously employed this strategy to improve vaccine efficiency against *T. gondii*, where we showed that a *T. gondii* antigen, SAG1, targeted to DCs via single-chain variable fragment (scFv) against DEC205 (NLDC-145) by intranasal and subcutaneous administration, improved protection against chronic *T. gondii* infection [29]. Interestingly, in a previous study, Michon et al. (2015) successfully expressed NLDC-145 scFv on the surface of *Lactobacillus plantarum* using different anchoring techniques [30]. Depending on the surface anchoring method, surface display of anti-DEC205 scFv efficiently enhanced *L. plantarum* internalization. Taken together, these findings suggest that anti-DEC205 scFv would be a suitable choice for assessing *T. gondii*’s capacity to produce functional membrane-anchored antibodies.

Therefore, to test our hypothesis, we sought to engineer a *T. gondii* strain displaying anti-DEC205 NLDC-145 scFv on its surface. The scFv comprises the heavy (VH) and light (VL) chains of an antibody linked with a flexible peptide [(Gly4Ser)3] in the VH-linker-VL orientation. A HA tag was incorporated in the N-terminus of the heavy chain for immune detection and characterization. Proteins displayed on the *T. gondii* tachyzoite’s surface are glycosylphosphatidylinositol (GPI)-anchored antigens known as SAGs (surface antigens). Among these surface proteins, SAG1, also referred to as SRS29B, is the most abundant antigen [31,32]. In this study, we investigated two possible strategies for anchoring anti-DEC scFv on the surface of *T. gondii*: direct GPI anchoring or fusion of SAG1 protein to the scFv sequence. Our results indicated that fusion of SAG1 protein resulted in a better functionality.

Next, to further explore if this approach can be expanded to construct other scFvs, we aimed to target tumor cells expressing the programmed death ligand 1 (PD-L1) by expressing anti-PD-L1 scFv on the surface of *T. gondii*. For this purpose, we used the scFv fragment of Atezolizumab, a monoclonal antibody against human PD-L1, which has been approved by the FDA for the treatment of advanced urothelial carcinoma and metastatic non-small cell lung cancer [33,34]. Structural studies of complexes formed between Atezolizumab and PD-L1 have revealed that all three complementary determining regions (CDRs) from the heavy chain of Atezolizumab are involved, whereas only two from the light chain (LCDR1 and LCDR3) form partial contacts [35,36]. In consideration of this hypothesis, inserting a HA tag coding sequence attached to the N-terminus region of the VH chain may impact the functionality of this fragment. Therefore, we compared a construct with the HA tag positioned outside the scFv, in the N-terminal part of SAG1 and linked via a GGGAS spacer to the C-terminus of the VL, to another construct where the HA tag is positioned at the N-terminal part of the VH. Our results indicate that indeed, in our model, the position of the HA tag may affect the interaction with PD-L1.

The aim of this study was to investigate in vitro, whether *T. gondii* engineered to express antibodies, in the scFv format displayed at the surface, enhances selective interaction with cells that express the antigen recognized by these scFvs. To the best of our knowledge, this is the first study to evaluate the functionality of antibodies expressed by *T. gondii* and to evaluate replicative *T. gondii* strains in tumor targeting.

## 2. Materials and Methods

### 2.1. Parasites

Tachyzoites of the RH strain of *T. gondii* were cultivated in Human Foreskin Fibroblasts (HFFs; ATCC CRL-1634) in Dulbecco’s Modified Eagle Medium 2 mM L-glutamine (DMEM, Gibco, Thermo Fisher Scientific, Waltham, MA, USA) with 10% Fetal Bovine Serum (FBS; Dutscher, Issy-les-Moulineaux, France), 50 U/mL penicillin, 50 µg/mL streptomycin (Gibco, Thermo Fisher Scientific) and 1% HEPES (Biowest, Nuaillé, France). Cells were cultured at 37 °C in 5% carbon dioxide (CO_2_) and infected with tachyzoites when they were 90% confluent. Tachyzoites were collected when they were freshly egressed.

### 2.2. Cell Lines

Human breast cancer cell lines MDA-MB-231 (kindly provided by Professor Emilie Allard-Vannier, EA 6295 Nanomedicaments et Nanosondes, University of Tours, Toursm, France) were cultured in DMEM 2 mM L-glutamine supplemented with 10% FBS, 50 U/mL penicillin and 50 µg/mL streptomycin, and 1% Non-Essential Amino Acid (NEAA, PAN-Biotech, Aidenbach, Germany).

Two murine melanoma cell lines were also used, B16F10 transfected with mCherry fluorescent protein (kindly provided by Dr. Mehdi Khaled, UMR 1299, Gustave Roussy Institute, Paris-Saclay University, Paris, France), and B16K1 (kindly provided by Dr. Laurent Gros, IRCM, INSERM, U896; University of Montpellier, Montpellier, France). These cells were grown in Roswell Park Memorial Institute medium 2 mM L-glutamine (RPMI 1640, Gibco, Thermo Fisher Scientific) containing 10% FBS and 50 U/mL penicillin and 50 µg/mL streptomycin.

The murine dendritic cell line MutuDC-1950 (kindly provided by Dr. Hans Acha-Orbea, University of Lausanne, Epalinges, Switzerland) was cultured in Iscove’s Modified Dulbecco’s Medium (IMDM, Gibco, Thermo Fisher Scientific) supplemented with 10% FBS, 50 U/mL penicillin and 50 µg/mL streptomycin, 10 mM HEPES and 50 µM β-mercaptoethanol (Sigma-Aldrich, St. Louis, MO, USA). The B3Z reporter hybridoma, specific for the MHCI-restricted peptide SIINFEKL (OVA257-264) in association with Kb class I MHC molecules, that express β-galactosidase under the control of the IL-2 promoter (kindly provided by Dr. Nicolas Blanchard, Center for Pathophysiology Toulouse-Purpan (CPTP), INSERM, CNRS, University of Toulouse, Toulouse, France) was grown in RPMI containing 2 mM L-glutamine, 10% FBS, 50 U/mL penicillin and 50 µg/mL streptomycin, 1% sodium pyruvate (Gibco, Thermo Fisher Scientific) and 50 µM β-mercaptoethanol. All cells were cultured at 37 °C in 5% CO_2_. For all experiments, cells were collected when they were in the logarithmic growth phase.

### 2.3. Antibodies Construct

Two expression cassettes were constructed to constitutively express proteins in *T. gondii* using 5′ αTub promoter and 3′ SAG1 UTRs. These sequences are available from GenBank, with accession numbers M20024 and X14080, respectively. One expression cassette was designed to express a CAT-GFP fusion protein to allow drug selection of stably transfected parasites [37,38], and the second was designed to express proteins of interest. In the latter, a five-repeat element was inserted in the *T. gondii* α-tubulin gene, upstream of the transcriptional start site (leading to promoter α-TUB8) for high-level expression of the protein of interest [39]. Two enzyme restriction sites, PmeI and NotI, were also included between the α-TUB8 and 3′UTR SAG1 sequences to allow insertion of the sequence encoding the protein of interest. The two cassettes (Appendix A) were cloned into pUC18, in the same orientation (pUC5) or in the reverse orientation (pUC8). In this study, all proteins of interest are membrane-bound proteins fused or not to the *T. gondii* surface antigen SAG1 (Genbank, accession number X14080), the major GPI-anchored protein of tachyzoites [40,41]. To achieve proper targeting, the proteins contain the following elements: the N-terminal signal sequence of SAG1 (MFPKAVRRAVTAGVFAAPTLMSFLRCGVMASD) including the Kozak sequence and the ATG start codon, the sequence encoding the scFv of interest, fused or not to the SAG1 sequence (D1D2 domains, [29]), the GPI anchor signal sequence of SAG1 (AAGTASHVSIFAMVIGLIGSIAACVA), and the stop codon.

The sequence encoding the scFv of interest comprised, a variable heavy chain (VH), a (GGGGS)x3 linker, a variable light chain (VL) derived from an anti-DEC205 (NLDC-145) [42] or an anti-PD-L1 (Atezolizumab, available on the IMGT information system) and a linker (GGGAS) at the C terminus. A HA tag (YPYDVPDYA) was incorporated at the N terminus of either the scFv or the SAG1 protein. The sequences for all newly constructed genes were confirmed by DNA sequencing (GATC Online).

### 2.4. Generation of Recombinant Toxoplasma gondii Strains

Transfections were performed with 10^7^ tachyzoites resuspended in cytomix buffer (120 mM KCl, 0.15 mM CaCl_2_, 10 mM K_2_ HPO_4_: KH_2_ PO_4_, 25 mM HEPES, 2 mM EDTA, 5 mM MgCl_2_, 3 mM ATP and 3 mM glutathione, pH 7.6) along with 50 μg of linearized plasmid (using a Biorad Gene PulserII Electroporator (Bio-Rad, Hercules, CA, USA) at settings of 2000 V, 50 ohms and 25 μF. Tachyzoites were then transferred to a fresh culture of HFF cells. After overnight growth, transfectants were selected with 20 μM chloramphenicol for three passages before cloning by limiting dilution in 96-well plates. Stably transfected clonal parasite lines were designated as follows: RH-DC2 (RH expressing an anti-DEC205 with N-terminal HA tag [DC2]), RH-DC2-SAG1 (RH expressing an anti-DEC205 fused to SAG1 with N-terminal HA tag [DC2-SAG1]), RH-PD-L1-N (RH expressing an anti-PD-L1 fused to SAG1 with N-terminal HA tag [anti-PD-L1-N]) and RH-PD-L1-C (RH expressing an anti-PD-L1 fused to SAG1 with C-terminal HA tag [anti-PD-L1-C]).

### 2.5. Recombinant DEC205 Protein (CF14)

The sequence coding for the ligand-binding activity of DEC205 (UniProt ID: Q60767), including the N-terminal cysteine-rich domain (CR), followed by the fibronectin type II domain (FN), and four C-type lectin-like domains (CTLD 1-4), was generated by gene synthesis (GeneArt, Regensburg, Germany). The synthetic gene was inserted in the plasmid vector pMT/BiP/V5 (Invitrogen, Waltham, MA, USA) using *Bgl*II and *Nhe*I restriction enzymes, and a Twin StrepTag (SAWSHPQFEK(GGGS)2GGSAWSHPQFEK) nucleotide-coding sequence was introduced at the C-terminal end using *Nhe*I and *Xho*I restriction enzymes. Schneider 2 cells (S2 cells) were transfected with the generated plasmid, using the Drosophila expression system (DES) purchased from Invitrogen. CF14 was purified from the supernatant of stably transfected cells, using a Strep-Tactin^®^ Sepharose^®^ column (2-1202-101, IBA, Göttingen, Germany) following the manufacturer’s protocol. The purified protein was analyzed by SDS-PAGE with Coomassie blue staining.

### 2.6. Analysis of HA-Tagged Proteins Expression

#### 2.6.1. ELISA Assay

ELISA was performed on whole tachyzoites of the recombinant *T. gondii* strains. In brief, 2 × 10^5^ parasites/well were coated on a flat-bottom 96-well plate (Maxisorp Nunc, Roskilde, Denmark) and fixed with 0.5% glutaraldehyde for 5 min at room temperature. The assay plates were blocked with phosphate-buffered saline (PBS) containing 4% Bovine Serum Albumin (BSA, Sigma) for 2 h at 37 °C. Following saturation, tachyzoites were incubated with rabbit anti-HA polyclonal antibody (Invitrogen) followed by goat anti-rabbit IgG, alkaline phosphatase conjugate (Sigma). Bound phosphatase activity was measured with p–nitrophenylphosphate (PNPP, Sigma) (1 mg/mL in DEA-HCl 1 M buffer, pH 9.8). The OD at 405 nm of each well was then read using a micro-plate reader (BioTek, Winooski, VT, USA).

#### 2.6.2. Immunoblot Analysis

Electrophoresis and immunoblotting of *T. gondii* tachyzoites were performed as previously described [43]. Briefly, freshly released tachyzoites (5 × 10^6^ tachyzoites/20 µL) were boiled in sodium dodecyl sulfate-polyacrylamide gel electrophoresis (SDS–PAGE) sample buffer containing dithiothreitol (50 mM) and loaded on homogenous 10% gels. After electrophoresis, proteins were transferred onto a nitrocellulose membrane and probed with rabbit anti-HA polyclonal antibody. followed by anti-rabbit secondary antibody conjugated to alkaline phosphatase (Sigma). Alkaline phosphatase activity was detected using the 5-bromo-4-chloro-3-indolyl phosphate/nitroblue tetrazolium (BCIP/NBT) liquid substrate system (Promega, Madison, WI, USA). The Prosieve QuadColor Protein Marker (Lonza, Basel, Switzerland) was used to determine the protein molecular weights.

#### 2.6.3. Dot Blot Assay

To compare the relative abundance of the HA-tagged proteins expressed in different recombinant *T. gondii* strains, 1 µL of 2-fold serially diluted tachyzoite crude lysates (prepared in SDS-PAGE sample buffer) was applied dot-wise to a dry nitrocellulose membrane. The membrane was then left to dry for 15 min before proceeding with the detection process using rabbit anti-HA polyclonal antibody followed by an anti-rabbit secondary antibody conjugated to horseradish peroxidase (Invitrogen), as described for immunoblot analysis. Dots were visualized using a luminol chemiluminescent substrate (SuperSignal West Pico PLUS Chemiluminescent Substrate, Thermo Fisher Scientific).

#### 2.6.4. Immunofluorescent Assay

For localization of the scFv to the surface of recombinant *T. gondii*, approximately 1 × 10^5^ freshly egressed tachyzoites were collected from infected HFF cells and washed in cold PBS. Tachyzoites were fixed in PBS containing 4% paraformaldehyde (PAF, Thermo Fisher Scientific) followed by incubation with rabbit anti-HA polyclonal antibody and a mouse anti-SAG1 (1E5) monoclonal antibody [44], simultaneously overnight at 4 °C. Following incubation, tachyzoites were double-stained with Alexa flour 488-conjugated goat anti-rabbit IgG and biotin goat anti-mouse IgG followed by Alexa flour 594-conjugated streptavidin for 30 min at room temperature. After washing in PBS, nuclei of tachyzoites were stained using Hoechst. Slides were mounted using Immu-Mount (Thermo Fisher Scientific), and images were captured with an Olympus IX73 fluorescent microscope using cellSens Dimension software Version 2.1. Except for the mouse anti-SAG1 monoclonal antibody, all antibodies and reagents were purchased from Thermo Fisher Scientific.

### 2.7. Characterization and Functionality of Recombinant T. gondii Expressing Murine Anti-DEC205 scFv

#### 2.7.1. Binding to DEC205 Protein

The functionality of the expressed anti-DEC205 (DC2 or DC2-SAG1) was tested by binding to the CF14. In an ELISA-based assay, Maxisorp 96-well plates were coated with 5 µg/mL of CF14 diluted in PBS and incubated overnight at 4 °C. After blocking of the coated wells with the saturation buffer for 2 h at 37 °C, tachyzoites of the selected RH-DC2 or RH-DC2-SAG1 clones were incubated with CF14 (5 × 10^5^/well) for another 2 h at 37 °C. Binding of the parasites to CF14 was assayed using *T. gondii* polyclonal antibody from infected rabbit serum followed by alkaline phosphatase-conjugated mouse monoclonal anti-rabbit IgG (Sigma). Bound phosphatase activity was revealed with 1 mg/mL PNPP (Sigma) and quantified by determining the absorbance at 405 nm using a micro-plate reader (BioTek).

#### 2.7.2. Binding to MutuDC Cells

Binding of the selected clones of RH-DC2 and RH-DC2-SAG1 to the murine dendritic cell line MutuDC-1950, which is known to express DEC205 [45], was determined by flow cytometry. MutuDC cells were harvested, and 10^6^ cells were mixed with 2 × 10^6^ parasites at a multiplicity of infection (MOI) of 2 in cold PBS containing 5% FBS and incubated for 1 h on ice. Unbound parasites were removed by three washes at 100 g for 5 min, then nonspecific binding was blocked by anti-FcγR monoclonal antibody (clone 2.4G2, eBioscience, San Diego, CA, USA). Parasite binding was detected by incubating cells with mouse monoclonal antibody T4 2EI2, specific for *T. gondii* tachyzoite surface glycoprotein, gp23 for 30 min on ice [31], followed by APC-conjugated anti-mouse IgG, for 30 min on ice. The fluorescence intensity of the cells was determined by flow cytometry (MACS Quant, Miltenyi Biotec, Bergisch Gladbach, Germany), and data were analyzed using FlowLogic software Version 7.2.1 (Miltenyi Biotech, Paris, France).

### 2.8. Characterization and Functionality of Recombinant T. gondii Expressing Human Anti-PD-L1 scFv

#### 2.8.1. Binding to Human PD-L1 Protein

To test the functionality of the expressed anti-PD-L1 by binding to human PD-L1, tachyzoites of the chosen clones were coated and fixed with glutaraldehyde on a flat-bottom P96-well plate. Assay plates were saturated for 2 h at 37 °C, then incubated with 10 µg/mL of a histidine-tagged recombinant human PD-L1 protein (Sino Biological, Beijing, China) for 1 h at 37 °C. Binding of PD-L1 protein to parasites was detected using mouse monoclonal anti-poly-histidine (Sigma) followed by alkaline phosphatase-conjugated anti-mouse IgG monoclonal antibody (Sigma). Bound phosphatase activity was revealed as previously described.

#### 2.8.2. Cell Stimulation and Flow Cytometry Analysis of PD-L1 Expression

The expression of the surface PD-L1 on tumor cell lines was detected by flow cytometry. For MDA-MB231 and HFF cell lines, the cells were collected, washed in cold PBS containing 5% FBS, and 5 × 10^5^ cells were stained with PE-labeled human anti-PD-L1 (Invitrogen) for 30 min on ice. PD-L1 overexpression on B16F10 cells was induced using recombinant murine interferon gamma (IFN-γ Gibco, Thermo Fisher Scientific). Briefly, 5 × 10^5^ B16F10 cells per well were seeded on a 6-well plate and incubated overnight to allow for adherence. Following adherence, cells were treated with 20 ng/mL of IFN-γ for 24 h. PD-L1 expression in IFN-γ-stimulated B16F10 cells as well as rested (non-stimulated) B16F10 and B16K1 cells was analyzed by staining 5 × 10^5^ cells with APC-conjugated murine anti-PD-L1 (Invitrogen) for 30 min on ice.

#### 2.8.3. Binding to PD-L1-Expressing Tumor Cells

Human cells, MDA-MB231 tumor cells and HFF cells and murine tumor cells, IFN-γ-stimulated B16F10, rested B16F10 and rested B16K1 cells (5 × 10^5^) were mixed with freshly collected tachyzoites of either RH-PD-L1-C or RH-DC2-SAG1 at MOI 5 for 2 h at 4 °C in complete cell medium. Following incubation, cells were washed at 100 g, 5 min to remove unbound parasites. Pellets were re-suspended in PBS and adhesion was measured by the percentage of GFP-positive cells using flow cytometry. For the neutralization assay, 5 × 10^5^ MDA-MB231 cells were incubated with a saturating concentration of Atezolizumab (20 µg/mL), and 5 × 10^5^ IFN-γ-stimulated B16F10 cells were incubated with increased concentrations of Atezolizumab (ranging from 1–1000 ng/mL) for 2 h at 4 °C. Following incubation with Atezolizumab, RH-DC2-SAG1 or RH-PD-L1-C tachyzoites were added for another 2 h. Adhesion was measured as previously described.

### 2.9. In Vitro Oncolytic Activity Assay

For the in vitro oncolytic activity experiments, MDA-MB-231 and IFN-γ-stimulated B16F10 cells were applied onto flat-bottomed 96-well culture plates at densities of 10^4^ cells/well and let overnight to attach. Cells were then either left uninfected or infected with RH wild-type or RH-PD-L1-C at MOI 3 and incubated for 24, 48 or 72 h. After each time point, medium was discarded and 100 µL of medium containing 10% of 3-(4,5-Dimethylthiazol-2-yl)-2,5-Diphenyltetrazolium Bromide reagent (MTT, Invitrogen) was added and incubated for another 4 h. The dark purple crystals that formed in intact cells were dissolved in DMSO, and viable cells were quantified by measuring the absorbance at 490 nm using a micro-plate reader (BioTek). Cell viability was calculated as a percentage of the control uninfected cells.

### 2.10. Mixed Cell Assay

A mixed-cell assay was performed using a mixture of targeted and non-targeted cells. For this purpose, IFN-γ-stimulated B16F10 (targeted) and rested B16K1 (non-targeted) cells were collected and mixed together at a ratio of 1:1. The cell mixture was incubated with RH-PD-L1-C at MOI 5 for 2 h at 4 °C. The cell mixture incubated with RH-DC2-SAG1 was used as a control. After washing at 100 g, 5 min to remove unbound parasites, cells were analyzed using flow cytometry. The two cell populations were distinguished by mCherry expression and the percentage of GFP-positive cells among each of the two cell groups indicated the percentage of parasite binding.

### 2.11. Co-Culture of Recombinant RH-PD-L1-C with B16F10 Tumor Cells and B3Z CD8+ T Cells

B16F10 cells were pretreated with recombinant IFN-γ (20 ng/mL) for 24 h, then washed with medium to remove IFN-γ before plating on 96-well culture plates at densities of 5 × 10^4^ cells/well. Cells were then incubated with 10 ng/mL of ovalbumin peptide MHC1 (OVA 257-264, Invivogen, San Diego, CA, USA) for 4 h at 37 °C. Following incubation, cells were washed with RPMI medium without red phenol to remove ovalbumin peptide and red phenol. Cell surface PD-1 expression on B3Z cells was confirmed by flow cytometer using FITC-conjugated mouse anti-PD-1 (Invitrogen). Subsequently, B3Z cells were co-cultured with B16F10 cells (ratio 1:1) alone or in the presence of the wild-type RH or RH-PD-L1-C recombinant tachyzoites at MOI 3 in medium without red phenol (total volume of 200 µL/well). These co-cultures were incubated for 16 h at 37 °C, and then 150 µL of supernatants were harvested and assessed for interleukin-2 (IL-2) using an ELISA kit (Invitrogen) according to manufacturer’s protocol. For measurement of β-galactosidase activity, cells mixtures were lysed by adding 50 µL of 0.01% Triton 100X (Sigma), and then 100 µL of β-galactosidase substrate [CRPG (Chlorophenol red-β-D-galactopyranoside), Roche, Basel, Switzerland] in HEPES 0.1 M was added to each well. Plates were incubated for 1 h at 37 °C and β-galactosidase activity was revealed by measuring absorbance at 570 nm.

### 2.12. Statistics

All analysis was performed using GraphPad Prism 8.00 Software. Due to the non-normal distribution of the data, statistical significance was analyzed by the non-parametric Mann–Whitney and Kruskal–Wallis tests. Estimation of the Gaussian distribution of the data was performed using Shapiro–Wilk and D’Agostino–Pearson normality tests. All the data are displayed as medians and interquartile ranges. Statistical differences are indicated with asterisks: * *p* < 0.05, ** *p* < 0.01.

## 3. Results

### 3.1. Design of Anti-Murine DEC-205 Single Chain Variable Fragment (scFv)

In our previous work by Lakhrif et al. [29], scFv directed against DEC205 was constructed by using the heavy (VH) and light (VL) chain variable regions of the monoclonal antibody NLDC145, in VH-linker-VL orientation. A sequence-encoding SAG1 protein was added to the C-terminus of the scFv via a GGGAS spacer. Moreover, a similar anti-DEC205 scFv construct was successfully engineered on the surface of *L. plantarum* by using different anchoring strategies to increase targeting and internalization in DCs [30]. In their construct, Michon et al. added a HA tag sequence in the N-terminus of the scFv for immune detection and characterization. Hereby, and in the light of this information, we hypothesized that *T. gondii* can be engineered to express surface anti-DEC205 scFv tagged with a HA tag in the N-terminus. The previously constructed recombinant scFv sequence [29] was utilized, preceded by the Kozak sequence, the ATG start codon, the sequence encoding the N-terminal signal sequence of SAG1, and a HA tag for immune detection. For attachment of the scFv to the tachyzoite’s membrane, a GPI anchor was used by adding a sequence encoding the SAG1 anchor signal (GPI). Using this skeleton, two anti-DEC205 configurations were developed (Figure 1A). In the first construct (DC2), the SAG1 anchor signal (GPI) with a stop codon was attached to the C-terminus of the scFv by a short spacer (GGGAS). In the second construct (DC2-SAG1), a sequence encoding SAG1 was introduced after the spacer (at the C-terminus of the scFv) followed by the SAG1 anchor signal (GPI) and the stop codon. Both sequences were flanked in 5′ by a PmeI site and in 3′ by a NotI site. Consequently, the sequences of DC2 and DC-SAG1 were cloned into pUC8 plasmid using PmeI and NotI sites to produce pUC8DC2GPI and pUC8DC2GPISAG1, respectively. Next, the plasmids were transfected into tachyzoites of *T. gondii* strain RH to generate the recombinant strains RH-DC2 and RH-DC2-SAG1, respectively.

### 3.2. RH-DC2 and RH-DC2-SAG1 Recombinant Tachyzoites Express Functional Anti-DEC205 scFv

A set of ELISAs were conducted to screen the surface display of the anti-DEC205 (DC2 and DC2-SAG1), in various chosen clones from RH-DC2 and RH-DC2-SAG1 strains, respectively. Tachyzoites from the selected clones were collected from infected HFF cells and incubated with anti-HA antibody. As shown in Figure 1B, all examined clones exhibited significant levels of expression compared to wild-type RH, with no discernible variation in the expression levels. For further investigations, one clone from each recombinant strain was chosen. To analyze if DC2 and DC2-SAG1 are correctly expressed by both engineered recombinant strains, whole-parasite cell lysates were analyzed by Western blotting. Wild-type RH was used as control. As revealed in Figure 1C, the DC2 and DC2-SAG1 proteins migrate at the expected size under reducing conditions (around 30 kDa and 55 kDa, respectively).

Next, the functionality of DC2 and DC2-SAG1 displayed at the surface of *T. gondii* was validated by their binding to the CF14. This recombinant mouse DEC205 protein, represented by the N-terminal part of the murine DEC205 receptor, which binds NLDC145 [46], was produced in the Drosophila Schneider 2 cell line. The ability of RH-DC2 and RH-DC2-SAG1 to recognize and bind to the recombinant DEC205 protein was assayed by ELISA (Figure 1D). Both engineered strains showed efficient binding to CF14, though RH-DC2-SAG1 exhibited increased binding compared to RH-DC2.

The feasibility of binding to targeted cells was also assessed using the murine dendritic cell line, MutuDC-1950. Tachyzoites of RH-DC2 and RH-DC2-SAG1 were incubated with MutuDC cells at 4 °C to allow parasite attachment to cells without internalization. Binding was demonstrated by immunostaining of bound parasites followed by flow cytometer analysis. The results were presented as percentages of binding (Figure 1E). Both RH-DC2 and RH-DC2-SAG1 demonstrated higher ability to bind to MutuDC compared to the wild-type strain. In agreement with the previous results, higher binding capacity was seen with RH-DC2-SAG1 than RH-DC2 (58% and 45%, respectively, compared to 20% for RH). On the basis that surface-expressed DC2-SAG1 showed higher binding to recombinant DEC205 protein and MutuDC cells than DC2, protein quantities expressed by RH-DC2-SAG1 and RH-DC2 were investigated. As a means of semi-quantification, dot blot analysis was performed using enhanced chemiluminescence (ECL) substrates (Figure 1F). Interestingly, in parasite cell lysates, RH-DC2 tended to show higher protein expression than RH-DC2-SAG1 (roughly three-fold).

Nonetheless, RH-DC2-SAG1 demonstrated the strongest binding to the targets. Taken together, these results suggest that fragments, vectorized in *T. gondii*, have been correctly expressed and were able to recognize DEC205 protein. Fusion of the SAG1 encoding sequence to the anti-DEC205 scFv resulted in enhanced functionality in terms of binding to targets.

### 3.3. Design of Human Anti-PD-L1 scFv

The results obtained from the successful surface display of anti-DEC205 were encouraging for broadening the uses of this innovative approach for further applications. Thus, targeting PD-L1-expressing tumors through surface display of anti-PD-L1 scFv was chosen to be investigated. For this purpose, the heavy- and light-chain variable regions of the human monoclonal antibody, Atezolizumab, connected with a flexible linker in VH-linker-VL orientation, were utilized. Furthermore, to investigate the impact of HA tag localization on the functionality of the scFv, two anti-PD-L1 configurations were developed (Figure 2A). In one construct, as for scFv anti-DEC205, the HA tag encoding sequence was incorporated in the N-terminus region of the VH chain (anti-PD-L1-N), while in the second construct it was inserted in the N-terminus of SAG1 (attached to the C-terminus of the VL chain by the GGGAS spacer) (anti-PD-L1-C). These sequences were then cloned into the plasmid pUC5 using PmeI and NotI cloning sites to generate pUC5PD-L1-N and pUC5PD-L1-C vectors, respectively. Subsequently, *T. gondii* strain RH tachyzoites were transfected with either of the two vectors to generate the recombinant strains RH-PD-L1-N and RH-PD-L1-C, respectively.

### 3.4. Anti-PD-L1 scFvs Are Correctly Expressed in T. gondii

Next, three clones from each recombinant strain were selected to assess anti-PD-L1 surface display in ELISA-based assays (Figure 2B). The results indicated that anti-PD-L1 scFv was markedly expressed in all the selected clones compared to the wild-type strain, which was used as control. The clone from each strain that displayed the highest expression (the highest OD measurement) was then chosen for the subsequent characterization and evaluation experiments. In order to confirm that the anti-PD-L1 scFv is correctly expressed at the expected size, whole-parasite cell lysates of each of the selected clones were analyzed by immunoblotting. Figure 2C demonstrates that both RH-PD-L1-N and RH-PD-L1-C were equally able to generate a molecule with an apparent size corresponding to a scFv linked to SAG1. The bands obtained were comparable to that obtained by RH-DC2-SAG1. However, a faint band (at around 27 kDa) was observed with the migration of anti-PD-L1-N.

### 3.5. RH-PD-L1-C but Not RH-PD-L1-N Strain Expresses a Functional Anti-PD-L1 scFv

To explore whether the membrane-expressed anti-PD-L1 scFv could bind to the immune checkpoint PD-L1, the binding of recombinant tachyzoites to a commercial recombinant human PD-L1 with a C-terminal polyhistidine tag (His tag) was analyzed by ELISA (Figure 2D). RH-PD-L1-C tachyzoites exhibited strong binding to the recombinant human PD-L1 indicating that the expressed surface anti-PD-L1 scFv can recognize and bind to its target. Conversely, RH-PD-L1-N tachyzoites could not demonstrate similar functionality since no binding to the recombinant human PD-L1 was observed. To verify this finding, a set of ELISA assays was performed to screen all the chosen clones of RH-PD-L1-C and RH-PD-L1-N for binding to the recombinant human PD-L1 (Appendix A). As per the earlier findings, all RH-PD-L1-C clones showed significant binding to PD-L1 protein; nonetheless, none of the RH-PD-L1-N clones showed any binding.

To further analyze these findings, dot blotting of whole-parasite cell lysates of RH-PD-L1-C and RH-PD-L1-N using ECL substrates was performed (Figure 2E). As the dot blot is revealing, the anti-PD-L1 scFv was adequately detected in both recombinant parasite cell lysates. Interestingly, the dot intensities indicate that anti-PD-L1 scFv was expressed in higher quantities in RH-PD-L1-N lysates. These findings imply that, despite the increased expression of anti-PD-L1 scFv by RH-PD-L1-N, when whole parasites were employed, only RH-PD-L1-C has been shown to be functional. Accordingly, we selected RH-PD-L1-C strain for further characterization and investigation.

### 3.6. Surface Display of the Anti-PD-L1-C scFv

In order to guide the expression of the anti-PD-L1 to the membrane of *T. gondii*, the sequence encoding the N-terminal sequence signal of SAG1 that guides the surface transport of the protein was incorporated. Additionally, to promote anchoring and fixation on the *T. gondii* membrane, the anchoring GPI sequence signal with the entire sequence of the mature SAG1 was also added [41]. To confirm that the anti-PD-L1 expression is localized in the *T. gondii* membrane as it is intended to be, an immuno-fluorescence microscopy was performed. To do so, RH-PD-L1-C tachyzoites were collected and the membrane was stained in red using mAb against SAG1, while the anti-PD-L1 fragment was stained in green by targeting the HA tag. As illustrated in Figure 2F, upon merging of the two images together, the orange color that results from the green–red merge is concentrated in the membrane, indicating clearly that the protein is successfully targeted to the parasite’s membrane.

### 3.7. RH-PD-L1-C Strain Exhibits Strong Binding to Human PD-L1-Expressing Cells In Vitro

Next, the aim was to investigate whether anti-PD-L1 display on the surface of *T. gondii* would direct the parasites to cells expressing PD-L1. The human breast cancer cell line, MDA-MB231, which is known to express PD-L1 on the cell membrane, was used, whereas HFF cells were used as PD-L1-negative control. RH-DC2-SAG1, as a recombinant, GFP-positive strain expressing an irrelevant surface scFv, was used as negative control. Surface PD-L1 display on MDA-MB231 and HFF cell lines was analyzed by flow cytometry (Figure 3A). This analysis verified that MDA-MB231 cells express substantial PD-L1 levels, while HFF cells were PD-L1-negative. Then, using HFF cells as non-targeted (PD-L1-negative) cells, RH-PD-L1-C or RH-DC2-SAG1 tachyzoites were incubated with either of the cells for 2 h at 4 °C. Using flow cytometer, cell binding was measured as a percentage of GFP-positive cells. As shown in Figure 3B, RH-PD-L1-C exhibited more pronounced binding to MDA-MB231 cells compared to the non-targeted, HFF cells (33% versus 7%, respectively). While, on the other hand, RH-DC2-SAG1 demonstrated comparable binding between the two cell lines regardless of the PD-L1 expression (5% and 7%). To verify that this enhanced binding of RH-PD-L1-C to MDA-MB231 cells was induced by PD-L1 expression; cells were incubated with saturating concentrations of Atezolizumab to block PD-L1 binding sites. Following saturation, RH-PD-L1-C or RH-DC2-SAG1 were added to MDA-MB231 cells, and binding was assessed by flow cytometry (Figure 3C). RH-PD-L1-C binding to MDA-MB231 cells was significantly decreased when PD-L1 was blocked with Atezolizumab, whereas RH-DC2-SAG1 binding remained unaffected, confirming that binding was in fact dependent on PD-L1 expression.

### 3.8. RH-PD-L1-C Strain Exhibits Strong Binding to Murine PD-L1-Expressing Cells In Vitro

Studies showed that the human anti-PD-L1, Atezolizumab, can efficiently bind to murine PD-L1 [47,48]. Subsequently, to test if the recombinant RH-PD-L1-C strain would bind to murine tumor cells expressing PD-L1, the murine melanoma cell lines B16K1 and B16F10 were employed. The basal PD-L1 expression on these cell lines, B16K1 and B16F10, is low to moderate, respectively. However, this moderate PD-L1 expression on B16F10 can be upregulated by stimulating cells with IFN-γ. First, the basal and induced expression of PD-L1 was confirmed on the rested (un-stimulated) B16K1 and B16F10 cells and IFN-γ-stimulated B16F10 cells by flow cytometry. As illustrated in Figure 4A, the three cell populations showed three different levels of PD-L1 expression. Subsequently, to evaluate the impact of different PD-L1 expression levels on binding, rested B16K1 and B16F10 cells and IFN-γ-stimulated B16F10 cells were incubated with RH-PD-L1-C or the control strain, RH-DC2-SAG1, for 2 h at 4 °C. Binding was analyzed by flow cytometry and represented by the percentages of GFP-positive cells (Figure 4B). Recapitulating the results obtained in a human cell line, the RH-PD-L1-C strain, similarly, exhibited increased binding to PD-L1-expressing murine tumor cells. Interestingly, the binding was shown to be increased in accordance with the increased PD-L1 expression levels, with the IFN-γ-stimulated B16F10 cells showing the highest level of binding.

Furthermore, to negate the possibility that this increased binding was induced by any other factors than PD-L1 expression, a neutralization assay was performed by incubating IFN-γ-stimulated B16F10 cells with increasing concentrations of Atezolizumab to block PD-L1 binding sites. The indicated parasites were then added for 2 h, and binding (GFP-positive cells) was analyzed by flow cytometer. As illustrated in Figure 4C, neutralization with Atezolizumab led to inhibition of RH-PD-L1-C binding to B16F10 cells in a concentration-dependent manner. Maximal inhibition was achieved with saturating concentrations of Atezolizumab (1 µg/mL), where binding of RH-PD-L1 was equivalent to the binding of the control, RH-DC2-SAG1. When taken together, these findings suggest that RH-PD-L1-C could efficiently bind to both human and murine tumor cells and this cell binding is dependent upon surface display of PD-L1.

### 3.9. RH-PD-L1-C Selectively Bind to Targeted Cells in Mixed Cell Population

Next, we asked if the recombinant RH-PD-L1-C strain would specifically target PD-L1-overexpressing cells when they are in a mixed-cell population. To answer this question, the targeted IFN-γ-stimulated B16F10 cells that also expressed mCherry fluorescent protein were mixed with non-targeted B16K1 cells at a ratio of 1:1. The cell mixture was then incubated for 2 h at 4 °C with either the RH-PD-L1-C or RH-DC2-SAG1 strain, and targeting was analyzed by flow cytometry. As shown in Figure 4D, the two cell populations in the cell mixture were distinguished by the mCherry expression (left dot blot). Then, binding was indicated according to the percentages of GFP-positive cells in each group of cells. The control strain, RH-DC2-SAG1, maintained almost 50–50% targeting (around 3% binding) on both cell groups with no respect for PD-L1 expression (middle dot blot). In contrast, RH-PD-L1-C tachyzoites predominantly bound to the PD-L1-overexpressing cells, with a negligible number of tachyzoites showing off-target binding (right dot blot).

### 3.10. Oncolytic Activity of RH-PD-L1-C In Vitro

To correlate the enhanced binding of the RH-PD-L1-C strain to tumor cells expressing PD-L1 to cellular toxicity, the infectious and oncolytic ability was compared to those of the wild-type strain in human (MDA-MB231) and murine (B16F10) tumor cell lines. MDA-MB231 and IFN-γ-stimulated B16F10 were infected with either RH wild-type or RH-PD-L1-C strain at MOI 3 or left uninfected, and cell viability was assessed after 24, 48 and 72 h. As shown in Figure 5A, results demonstrated that RH-PD-L1-C could efficiently infect and kill both human and murine tumor cells with a significantly higher capacity than the parental *T. gondii* strain.

### 3.11. RH-PD-L1-C Strain Inhibits the Binding of PD-1/PD-L1 and Promotes T Cell Activation

The ability of the anti-PD-L1 scFv expressed on RH-PD-L1-C tachyzoite surface to interfere with PD-1/PD-L1 interaction and to promote T-cell responses was evaluated in in vitro co-culture assay. Murine B16F10 tumor cells incubated with MHCI-restricted peptide SIINFEKL (OVA257-264) were used as target cells. Murine B3Z CD8+ T cells, which recognized the SIINFEKL bound to H-2Kb MHC-I and OVA-peptide, were used as antigen-specific CD8+ effector T cells. Activated B3Z cells also express β-galactosidase under the control of the IL-2 promoter; thus, the activity of B3Z cells can be measured by β-galactosidase and IL-2 expression. For the co-culture assay, first, surface PD-1 display on B3Z was confirmed by flow cytometry (Figure 5B). Then, B16F10 cells were stimulated to over-express PD-L1 by IFN-γ before co-culturing with B3Z in the presence of RH-PD-L1-C or RH wild-type (control) tachyzoites. B3Z cells’ response (measured as β-galactosidase activity) to B16F10 was significantly enhanced in the presence of RH-PD-L1-C tachyzoites in respect to wild-type RH (Figure 5C). To obtain further evidence, IL-2 secretion was measured. As illustrated in Figure 5D, the RH-PD-L1-C strain was able to stimulate more IL-2 secretion compared to RH wild-type. These findings indicate that surface display of anti-PD-L1 on the surface of *T. gondii* not only increases the targeting of PD-L1-positive cells but may also contribute to blocking the PD-1/PD-L1 pathway.

## 4. Discussion

We hereby present, for the first time, the generation and characterization of replicative *Toxoplasma gondii* strains displaying surface single-chain variable fragments as a novel strategy for targeting specific cells. The strategic use of *T. gondii* in the context of several diseases offers a promising therapeutic approach. Given *T. gondii*’s ability to invade all nucleated cells, one approach to reduce nonspecific cell invasion and promote selectivity is by targeting parasites to receptors that are unique or overexpressed by the targeted cells [49]. Therefore, we aimed to determine whether engineering *T. gondii* to express surface antibody fragments would induce specific cell targeting. To test this hypothesis, we selected Dendritic Cells (DCs) as the target by employing a single-chain variable fragment (scFv) against the murine DC-restricted antigen-uptake receptor DEC205. Our research group [29] and others [50] investigated DC targeting as a strategy to deliver vaccines to DCs, thereby enhancing antigen presentation and improving the immune response. Previously, Michon et al. demonstrated the feasibility of targeting DCs through membrane display of anti-DEC205 scFv on the surface of recombinant *Lactobacillus plantarum* strains [30]. Building upon this approach, we hypothesized that we could engineer *T. gondii* to express anti-DEC205 scFv on its surface. Hereby, we report the generation and characterization of recombinant *T. gondii* strains displaying surface mouse anti-DEC205 scFv. Two strategies were tested to generate *T. gondii* stable transfectants expressing surface scFv: direct GPI anchoring or GPI anchoring in addition to fusion of SAG1 protein to the scFv sequence. Subsequently, two recombinant strains were generated, named RH-DC2 and RH-DC2-SAG1, respectively. In previous studies, a few heterologous proteins have been expressed on the surface of *T. gondii*. Anchoring the protein of interest by GPI has been successfully achieved for both alkaline phosphatase and Cryptosporidium Glycoprotein Antigen Cpgp40/15 [51,52]. However, membrane display of heterologous surface proteins by *T. gondii* through fusion with SAG1 has been reported to be unsuccessful, resulting in either undetectable or short truncated protein products expressed in the cytoplasm of recombinant parasites, as observed for the *Plasmodium yoelii* circumsporozoite protein [53]. Furthermore, in a study by Gregg et al. aimed at targeting Ovalbumin to various *T. gondii* oragnelles, GPI-membrane-anchored Ovalbumin was indeed targeted to the surface; however, unstable expression was observed, appearing as degraded bands on Western blot analysis with some amount being shed in the culture supernatant. In contrast to these findings, our results indicate that both strategies were effective in expressing stable membrane scFv, as illustrated in the immune-blotting figures. Furthermore, our dot blot analysis data demonstrated higher quantities of the anti-DEC205 scFv detected in RH-DC2 parasite lysates. Nonetheless, in subsequent functionality assays for binding to the recombinant DEC205 protein using intact tachyzoites, RH-DC2-SAG1 showed higher binding ability than RH-DC2. The lower binding capacity of RH-DC2, despite the protein being highly expressed, is most likely due to limited accessibility to the receptor, potentially resulting from its structural assembly.

*T. gondii* possesses a natural ability to bind to and invade all nucleated mammalian cells. A previous work by Swee et al. (2015) demonstrated that the addition of a single-domain antibody against CD19 to the *T. gondii* surface via sortagging significantly increased parasite targeting to B cells while decreasing binding to non-B cells [54]. Although effective for targeting, sortagging requires the production of recombinant proteins, conjugation for targeted tachyzoites, and purification steps. Surface engineering of *T. gondii* offers a solution to these challenges. Thus, our study aimed to investigate whether surface display of an antibody against the DC receptor DEC205 would enhance parasite targeting to these cells compared to the parental parasite. To address this question, we utilized a DC cell line known to express DEC205 on its surface. Consistent with our hypothesis, the adhesion assays showed that the surface display of anti-DEC205 increased the binding of RH-DC2-SAG1 and, to a lesser extent, RH-DC2, to DCs. Taken together, these findings suggest that *T. gondii* can be successfully engineered to express functional surface scFv, and this surface display results in increased binding to the targeted cells compared to the wild-type strain. When the SAG1 encoding sequence was fused to the anti-DEC205 scFv, this construct exhibited enhanced functionality in terms of binding to targets.

These findings offer the possibility of further exploration for surface engineering applications of *T. gondii*. Therefore, we aimed to investigate the potential of this novel approach for tumor targeting. Generally, the reprogramming of oncolytic microorganisms to target tumor cells while sparing normal cells has been widely researched. One efficient way to generate tumor-specific microorganisms is by targeting them to cell surface receptors or tumor-associated antigens specifically expressed or significantly overexpressed on tumor cells. This approach has been successfully applied to various oncolytic bacteria and viruses through surface modification via the attachment of antibody fragments targeting various tumor receptors or antigens. Various targets have been explored, with HER2 [55,56] and CD20 [57,58] being among the most investigated targets.

In this study, we investigated the possibility of targeting programmed cell death ligand 1 (PD-L1). PD-L1 is an immune checkpoint protein that is overexpressed on many solid tumors [59]. This ligand interacts with the receptor-programmed cell death-1 (PD-1), expressed on the cell surface of T cells, B cells, monocytes, and natural killer (NK) T cells [60]. Activation of the PD-1/PD-L1 pathway negatively regulates T-cell-mediated immune responses and induces T-cell exhaustion. Several solid tumors upregulate PD-L1 expression as an immune evasion mechanism, making the inducible PD-L1 expression at the tumor site a selective target for antitumor therapy [61]. Furthermore, earlier studies on mice-bearing melanoma have demonstrated that treatment with *T. gondii* increases PD-L1 upregulation, rendering these tumor cells more sensitive for PD-L1 [23]. Targeting oncolytic microorganisms to PD-L1-positive cells has been described recently, primarily through modification of oncolytic viruses. This modification involves either using bispecific antibodies as adaptors between the adenovirus and the cell surface receptor [62], or coating the virus with bioengineered cell membrane nanovesicles expressing programmed cell death protein 1 (PD-1) [63].

For the construction of the anti-PD-L1 single-chain variable fragment (scFv), we utilized the heavy- (VH) and light- (VL) chain variable regions of the commercially available human anti-PD-L1 antibody, Atezolizumab. Building upon the successful fusion of the membrane protein SAG1 to the scFv in our previous anti-DEC205 constructs, we employed the same design approach for the anti-PD-L1 scFv. In the effort to generate a recombinant *T. gondii* expressing functional anti-PD-L1 scFv, we searched the scientific literature for characterized PD-L1 antibody fragments (scFvs) to select the most appropriate design and insertion sites of the HA tag. Emerging evidence indicates that, for binding of anti-PD-L1 monoclonal antibodies, the VH is essential, rather than VL chains [35], suggesting that the HA tag at the N-terminus of VH may negatively impact the functionality of the expressed fragment. However, in contrast to this theory, others have constructed a functional anti-PD-L1 scFv with the GFP-encoding sequence at the N-terminus of the VH [64]. Thus, we examined both strategies by constructing two anti-PD-L1 scFv configurations: one with the HA tag placed at the N-terminus of the VH chain (anti-PD-L1-N) and the other with the HA tag positioned outside the scFv, at the N-terminus of SAG1 (anti-PD-L1-C). Consequently, two recombinant strains were generated, named RH-PD-L1-N and RH-PD-L1-C, respectively. After confirming the display of anti-PD-L1 on the surface of both recombinant strains by ELISA and Western blotting, functionality was assessed by binding to recombinant PD-L1 protein. The strain engineered with the anti-PD-L1 scFv featuring the HA tag at the C-terminus exhibited robust binding to PD-L1 protein. Conversely, the strain engineered with the HA tag at the N-terminus of the scFv failed to recognize and bind to PD-L1, despite adequate protein expression. Consistent with the observations of Zhang et al. [35] our data indicate that positioning the HA tag at the very N-terminus of the scFv might interfere with its binding to PD-L1 protein, validating the crucial role of the VH chain in binding to PD-L1 protein.

We further evaluated the ability of RH-PD-L1-C to bind to tumor cells expressing PD-L1 by flow cytometry. For this adhesion analysis, we utilized the human breast cancer cell line MDA-MB-231 which is known to express high levels of PD-L1. In this analysis, we relied on GFP expression by the recombinant parasites where binding was indicated by the percentage of GFP-positive cells. As a control, we used the recombinant parasite RH-DC2-SAG1, which is also surface-modified, expressing an irrelevant scFv. Firstly, we confirmed that both surface-modified strains demonstrated low replication compared to the control in non-targeted HFF cells (Appendix A). Similarly, in the previous study by Michon et al. (2015) on Lactobacillus displaying surface anti-DEC205 scFv, they showed that the recombinant strains had a substantially lower growth rate than the wild-type bacteria; still, all strains showed reasonable growth [30]. However, when targeted PD-L1 was expressed on the cell surface, significantly higher binding was observed with RH-PD-L1-C to MDA-MB-231 cells. To further verify that this increased binding was induced by PD-L1/anti-PD-L1 interaction, a neutralization assay was performed. The neutralization assay with Atezolizumab confirmed that this binding was due to the specific interaction of RH-PD-L1-C with PD-L1 expressed by MDA-MB-231 cells.

In a study by Magiera-Mularz et al., the authors demonstrated that PD-L1 of both mouse and human origins share high structural similarity [48]. They indicated that Atezolizumab can bind to mouse PD-L1, and mice bearing MC38 cells were successfully treated with Atezolizumab. Based on this study, we theorized that our recombinant *T. gondii* expressing surface anti-PD-L1 scFv derived from Atezolizumab would bind to murine PD-L1-expressing tumor cells. To test this theory, we employed murine tumor cell lines, B16K1 and B16F10, which show low and moderate levels of PD-L1, respectively. It is well known now that PD-L1 expression can be upregulated by the IFN-γ receptor II signaling pathway. Therefore, by stimulating B16F10 cells with recombinant IFN-γ, we generated three levels of PD-L1 expression, low (B16K1), medium (B16F10) and high (IFN-γ-stimulated B16F10). The ligand–receptor interaction is generally influenced by the affinity of the ligand for the receptor and the expression and surface accessibility of the receptor. As expected, the surface-engineered *T. gondii* bound to cells in the order IFN-γ-stimulated B16F10 > B16F10 > B16K1, showing a strong correlation between binding and PD-L1 expression levels. This correlation was further verified by neutralizing PD-L1 binding sites after it had been upregulated with IFN-γ with increasing concentrations of Atezolizumab. The inhibition of binding by Atezolizumab in a concentration-dependent manner confirmed that the adhesion is mainly achieved via the interaction between scFv on the *T. gondii* surface and its ligand on the cell surface, rather than via the nonspecific interactions.

Thus far, we have confirmed that the surface display of anti-PD-L1 by *T. gondii* resulted in a high capacity to bind both human and murine PD-L1-positive tumor cells. However, these recombinant strains retain the capability to infect and replicate in non-targeted cells. Here, we investigated whether these recombinant parasites would specifically target and bind to PD-L1-overexpressing cells within a mixed-cell population. To address this assumption, we studied targeting in a cellular mixture consisting of PD-L1-overexpressing (IFN-γ-stimulated B16F10) cells and cells expressing low levels of PD-L1 (B16K1 cells). Our results showed that binding was primarily restricted to the cells that overexpressed the surface marker, with very few tachyzoites exhibiting off-target binding. Such data have been achieved before by using retargeted viruses or bacteria where invasion genes are deleted. In our model, we achieved targeting by a replicative strain. Selective targeting according to surface display of markers normally associated with diminished nonspecific cell targeting is of relevance to potential therapeutic applications.

In addition to immune system induction, the direct oncolytic activity achieved through intracellular multiplication within cancer cells is an argument in favor of using replicative rather than attenuated *T. gondii* strains. Therefore, we evaluated the capacity of our model to induce direct tumor cell death by invading and infecting PD-L1-positive tumor cells. Our results confirmed the superior killing ability of the recombinant replicative strain, RH-PD-L1-C, compared to the parental RH strain. These findings suggest that RH-PD-L1-C is capable of directly eradicating tumor cells in vitro in the absence of immune system influence, by infecting cells and causing their death.

PD-L1 expressed on tumor cells interacts with PD-1 on effector T lymphocytes resulted in T-cell exhaustion. Exhausted CD8 T cells lose their effector function, evidenced by their inability to secrete pro-inflammatory cytokines, such as IL-2, interferon gamma (IFN-g), and tumor necrosis factor alpha (TNF-a) [60,65]. Therefore, we evaluated the ability of our model to inhibit PD-1/PD-L1 interaction in a co-culture in vitro system of B16F10 tumor cells and B3Z CD8+ T cells to mimic a tumor microenvironment where T-cell activity will be inhibited by tumor cells. Our results showed that the addition of RH-PD-L1-C recombinant tachyzoites to the co-culture potentiated the T-cell activity measured by β-galactosidase and IL-2 secretion (by 1.7 and 1.6 fold, respectively) compared to the wild-type strain. These results suggest that anti-PD-L1 scFv expressed at the surface of *T. gondii* tachyzoites could achieve a combination of immune checkpoint blockade and efficient targeting of tumor cells. In addition, upregulation of PD-L1 by *T. gondii* would improve the efficacy of this strategy by presenting more targets. Interestingly, in a similar approach, adenoviruses coated in a bioengineered cell membrane expressing PD-1 to target PD-L1 were shown to effectively reactivate exhausted tumor-specific CD8+ T cells, inhibit tumor cell proliferation, and improve the anti-tumor efficacy in vitro and in vivo [63]. As a next step, future studies should be conducted to evaluate these recombinant tachyzoites in 3D cell culture systems and appropriate in vivo animal models to confirm their ability to target PD-L1+ tumor cells, leading to better antitumor activity.

## 5. Perspectives

The results presented here demonstrate that displaying antibody fragments on the surface of *T. gondii* effectively induces a targeted effect. Our strategy paves the way for the use of replicative strains of *T. gondii* that potentiate tumor oncolysis while localizing the therapeutic effect at the target site. This approach is promising in the context of various tumors.

## Figures and Tables

**Figure 1 cells-13-00975-f001:**
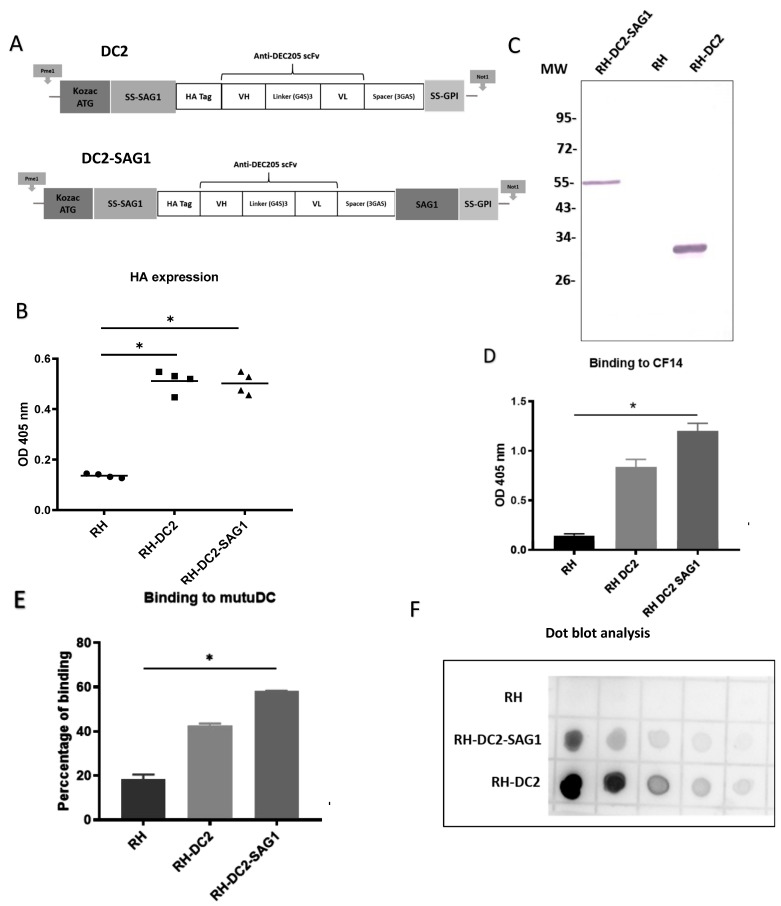
Engineering and characterization of recombinant *T. gondii* expressing a single-chain variable fragment directed against murine DEC205 (anti-DEC205 scFv). (**A**) Schematic representation of the anti-murine DEC205 with N-terminal HA-tagged scFv (DC2) and the anti-murine DEC205 fused to SAG1 with N-terminal HA-tagged scFv (DC2-SAG1). The locations of the Kozak sequence and start codon (ATG), signal sequences of the N-terminus surface antigen 1 (SS-SAG1) and glycosylphosphatidylinositol (GPI), human influenza hemagglutinin (HA) tag, antibody variable domains (heavy [VH] and light [VL]), peptide linkers ([Gly4Ser]3 and [Gly3AlaSer]), and surface antigen 1 sequence (SAG1) are indicated. (**B**) Screening based on HA expression by ELISA of different selected clones of RH-DC2 and RH-DC2-SAG1, using rabbit anti-HA followed by AP-conjugated anti-rabbit IgG. RH tachyzoites were used as a control (n = 3 replicates). (**C**) Western blot of parasite cell lysates of RH-DC2-SAG1 (line 1), RH wild-type (line 2), and RH-DC2 (Line 3) revealed using rabbit anti-HA followed by AP-conjugated anti-rabbit IgG. MW: molecular weight marker in kilo Dalton. (**D**) Binding of RH, RH-DC2, and RH-DC2-SAG1 tachyzoites to CF14 recombinant protein assessed by ELISA using *T. gondii* antibody from infected rabbit serum (n = 3 replicates). (**E**) Binding of RH (control), RH-DC2, and RH-DC2-SAG1 tachyzoites to murine dendritic cells MutuDC expressing DEC205 assessed by flow cytometer using a mouse monoclonal antibody specific for *T. gondii* gp23 glycoprotein, followed by APC-conjugated anti-mouse IgG. Binding was measured as the percentage of GFP-positive cells (n = 4 replicates). (**F**) Dot blot to compare the relative abundance of the HA-tagged proteins expressed by RH-DC2 and RH-DC2-SAG1. Serial 2-fold dilutions of tachyzoite crude lysates were loaded onto nitrocellulose membrane and probed with rabbit anti-HA followed by HRP-conjugated anti-rabbit IgG and detection with chemiluminescent HRP substrate. Statistical significance is indicated by * *p* < 0.05. In all ELISA experiments, optical densities were read at 405 nm. All experiments were repeated at least three times. AP: Alkaline Phosphatase. HRP: Horseradish peroxidase.

**Figure 2 cells-13-00975-f002:**
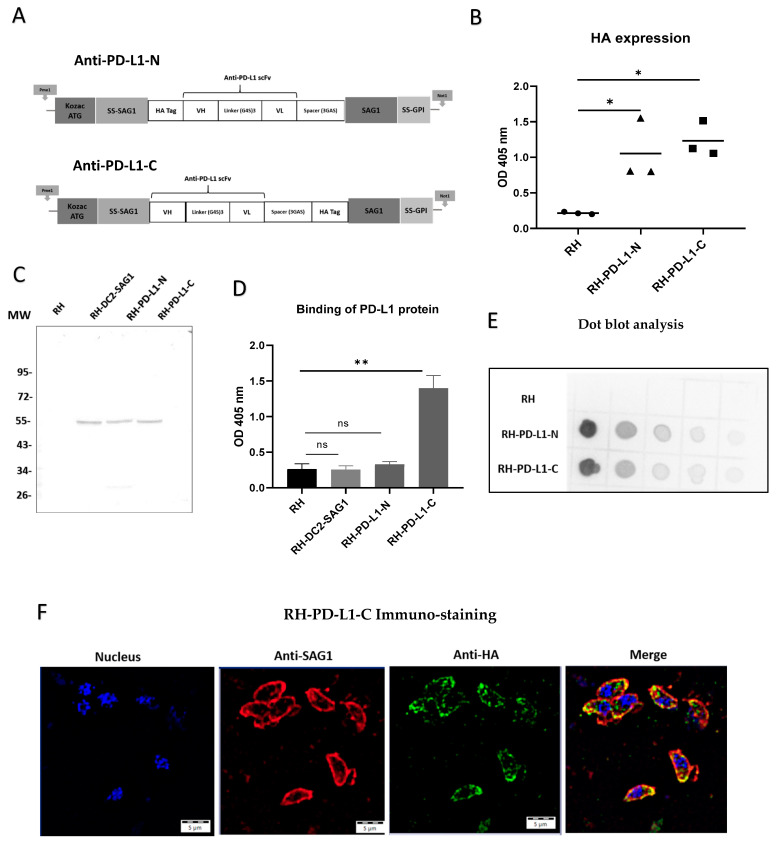
Engineering and characterization of recombinant *T. gondii* expressing a single-chain variable fragment directed against the human programmed death-ligand 1 (anti-PD-L1 scFv). (**A**) Schematic representation of the anti-human PD-L1 scFv with HA tag in the N-terminus (anti-PD-L1-N) and the anti-human PD-L1 scFv with HA tag in the C-terminus (anti-PD-L1-C). The locations of the Kozac sequence and start codon (ATG), signal sequences of N-terminus of surface antigen 1 (SS-SAG1) and of glycosylphosphatidylinositol (GPI), human influenza hemagglutinin (HA) tag, antibody variable domains (heavy [VH] and light [VL]), peptide linkers ([Gly4Ser]3 and [Gly3AlaSer]) and surface antigen 1 sequence (SAG1) are indicated. (**B**) HA expression in three selected clones of RH-PD-L1-N and RH-PD-L1-C analyzed by ELISA using rabbit HA polyclonal antibody followed by anti-rabbit IgG AP-conjugated. RH tachyzoites were used as control (n = 3 replicates). (**C**) Western blot of parasite cell lysates RH (Line 1), RH-DC-SAG1 (line 2), RH-PD-L1-N (line 3), and RH-PD-L1-C (line 4) revealed using rabbit anti-HA followed by anti-rabbit IgG AP-conjugated. MW: molecular weight marker in kilo Dalton. (**D**) Binding of histidine-tagged PD-L1 recombinant protein to RH-PD-L1-N and RH-PD-L1-C tachyzoites assessed by ELISA using anti-His tag antibody. RH and RH-DC2-SAG1 tachyzoites were used as control (n = 5 replicates). (**E**) Dot blot to compare the relative abundance of the HA-tagged proteins expressed by RH-PD-L1-C and RH-PD-L1-N tachyzoites. Serial 2-fold dilutions of tachyzoite crude lysates were loaded onto nitrocellulose membrane and probed with rabbit anti-HA followed by HRP-conjugated anti-rabbit IgG and detection with chemiluminescent HRP substrate. (**F**) Localization of anti-PD-L1 scFv display on the membrane of *T. gondii*. Free tachyzoites of RH-PD-L1-C recombinant strain were incubated with mouse SAG1 mAb followed by biotin anti-mouse IgG and AF-594-conjugated streptavidin (red) or rabbit anti-HA followed by anti-rabbit IgG-AF-488 (green). Nucleus was stained using Hoechst (blue). Images were captured at x600 with Olympus IX73 fluorescent microscope. Scale bars represent 5 μm. Statistical significance is indicated by * *p* < 0.05, ** *p* < 0.01. ns: not significant. In all ELISA experiments, optical densities were read at 405 nm. All experiments were repeated at least twice. AF: Alexa Fluor.

**Figure 3 cells-13-00975-f003:**
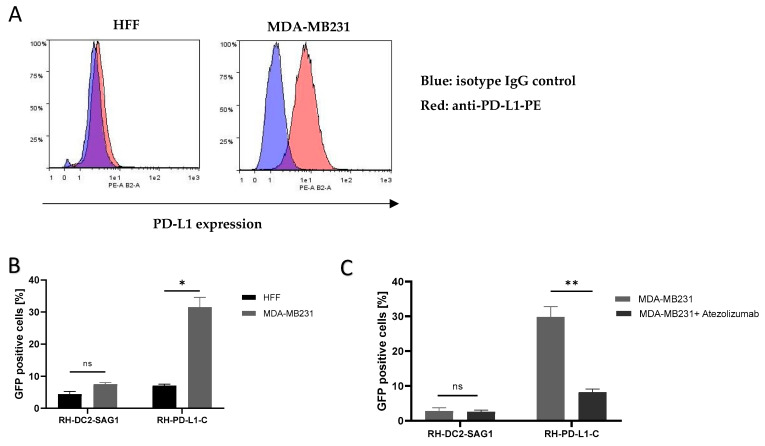
Binding of RH-PD-L1-C and RH-DC2-SAG1 (irrelevant scFv control) recombinant tachyzoites to human cells expressing PD-L1. (**A**) PD-L1 expression on HFF and MDA-MB231 cells. Cells were stained with PE-conjugated anti-human PD-L1 and analyzed by flow cytometry. Blue area depicts the profile of cells stained with isotype control IgG, and red area depicts the profile of cells stained with anti-PD-L1. (**B**) Quantification of binding of *T. gondii* recombinant tachyzoites to HFF or MDA-MB-231 cells analyzed by flow cytometry. Human cells were incubated with either RH-DC2-SAG1 or RH-PD-L1-C at MOI 5 and binding was indicated by percentages of GFP-positive cells (n = 3 replicates). (**C**) Binding competition with Atezolizumab. After incubating MDA-MB-231 cells with a saturating concentration of Atezolizumab, followed by incubation of the RH-PD-L1-C or RH-DC2-SAG1 strain, binding was quantified by flow cytometry (n = 5 replicates). Statistical significance is indicated by * *p* < 0.05, ** *p* < 0.01, ns: not significant. All experiments were repeated at least twice. MOI: multiplicity of infection.

**Figure 4 cells-13-00975-f004:**
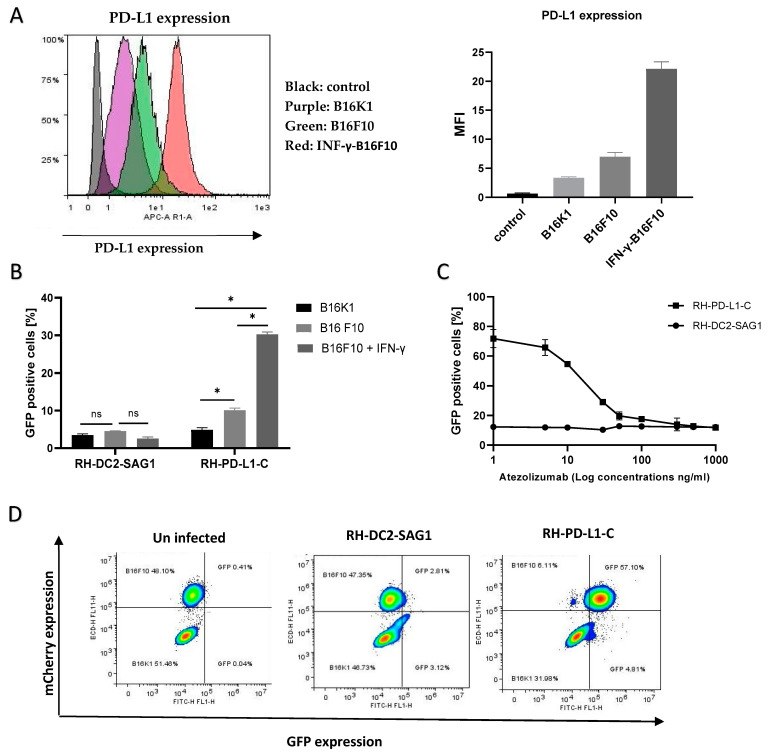
Binding of RH-PD-L1-C and RH-DC2-SAG1 (irrelevant scFv control) recombinant tachyzoites to murine tumor cell lines expressing PD-L1. (**A**) Surface display level of PD-L1 on IFN-γ-stimulated B16F10 (B16F10 + IFN- γ), non-stimulated B16F10 (B16F10) and B16K1 cells. Cells were stained with APC-conjugated mouse anti-PD-L1 before flow cytometry analysis. Results are represented as flow cytometer histograms (on the left) and mean fluorescence intensity values (MFI, on the right). Black area indicates the profile of cells incubated with isotype control IgG. Profile of cells stained with anti-PD-L1 are indicated in purple area (B16K1), green area (B16F10) and red area (IFN-γ-stimulated B16F10). (**B**) Murine cell lines were incubated with either RH-DC2-SAG1 or RH-PD-L1-C tachyzoites at MOI 5, and the percentage of GFP-positive cells was quantified by flow cytometer (n = 3 replicates). (**C**) Competitive binding assay. IFN-γ- stimulated B16F10 cells were incubated with increased concentrations of Atezolizumab followed by addition of either RH-DC2-SAG1 or RH-PD-L1-C tachyzoites at MOI of 5. Binding was assessed as a percentage of GFP-positive cells by flow cytometry (n = 3 replicates). (**D**) Selective targeting of PD-L1 over-expressing cells by RH-PD-L1-C in mixed-cell assay. B16K1 and IFN-γ-stimulated B16F10 cells were mixed together at a ratio of 1:1, and the cell mixture was incubated with RH-PD-L1-C or the control strain, RH-DC2-SAG1, at MOI of 5. The two cell populations were discriminated by mCherry expression by gating on mCherry-positive (B16F10) and mCherry-negative (B16K1) cells. Binding was indicated by the GFP-positive cells among each group. Statistical significance is indicated by * *p* < 0.05, ns: not significant. All experiments were repeated at least twice.

**Figure 5 cells-13-00975-f005:**
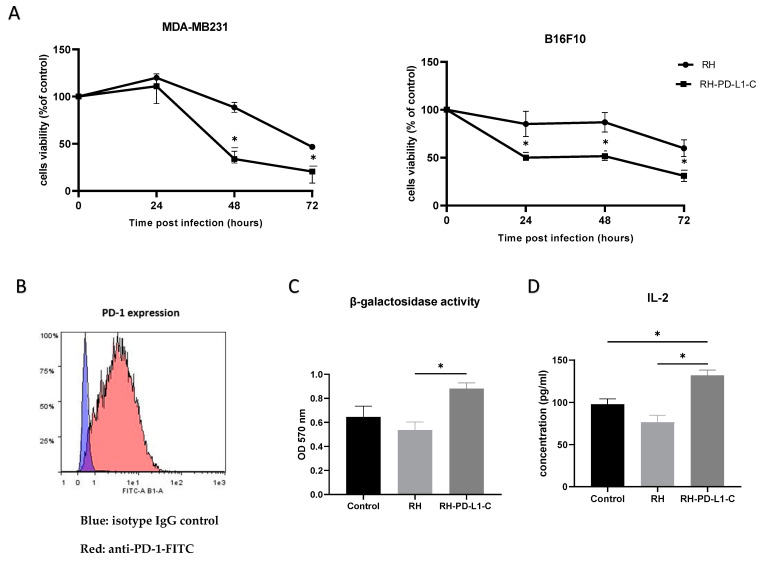
In vitro biological activity. (**A**) In vitro oncolytic activity. MDA-MB231 and IFN-γ-stimulated B16F10 cells were left untreated or infected with either the wild-type *T. gondii* RH strain or RH-PD-L1-C recombinant strain at MOI of 3. Cell viability was measured 24, 48 and 72 h post infection using MTT test. Results are represented as percentages of viable cells in respect to the control uninfected cells. Optical densities were read at 490 nm. Asterisks indicate statistically significant differences between RH wild-type-treated and RH-PD-L1-C-treated cells (* *p* < 0.01). All experiments were repeated at least twice. (**B**) PD-1 expression on B3Z cells was analyzed by flow cytometer using FITC-labeled mouse anti-PD-1. Blue area indicates the profile of cells incubated with isotype control IgG, while profile of cells stained with anti-PD-1 is indicated in red area. (**C**,**D**) Activation of the MHC class I-restricted T-cell line B3Z in the context of presentation of the SIINFEKL OVA peptide or presentation of the soluble OVA peptide SIINFEKL to B3Z cells. B3Z cells were co-cultured for 20 h with IFNγ-stimulated B16F10 cells loaded with OVA peptide in presence of RH (control) or RH-PD-L1-C recombinant tachyzoites. The B3Z response was measured as β-galactosidase activity (**C**) by measuring absorbance at 570 nm after incubation with β-galactosidase substrate (CRPG) and IL-2 quantification (**D**) in co-culture supernatants by ELISA. Statistical significance is indicated by * *p* < 0.05. All experiments were repeated at least twice.

## Data Availability

Data are contained within the article and Appendix A.

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
