# Peer review of "Specific Cell Targeting by *Toxoplasma gondii* Displaying Functional Single-Chain Variable Fragment as a Novel Strategy; A Proof of Principle"

_cells, 2024, doi:10.3390/cells13110975_

Round 1
Reviewer 1 Report
Comments and Suggestions for Authors
The study entitled “Specific Cell Targeting by Toxoplasma gondii displaying Functional Single Chain Fragment Variable as a Novel Strategy; a Proof of Principle”, conducted by the Muna et al., highlighted T. gondii specificity by surface-expression of scFv directed against Dendritic cells endocytic receptor, DEC205 and immune checkpoint PD-L1. This study is interesting for researchers working on control strategies against Toxoplasmosis and could provide efficient way toward control this infection. I recommend this MS for publication following minor corrections.
Line 35. Re-write the sentence.
Line 83 remove the space bar “We-previously-applied”
Line 427 merge two paragraphs
Line 473 use back space to merge two paragraphs into one
Line 488 use full stop “.” at the end of paragraphs.
Comments on the Quality of English Language
Minor editing is required
Reviewer 2 Report
Comments and Suggestions for Authors
Summary
This study designed a novel cell targeting system by engineering Toxoplasma gondii with scfv. Fusing SAG1 with Anti-DEC205 scFv, they created an anti-DEC-SAG1 scFv which can bind with DEC205-expressing MutuDC cells. Furthermore, they replaced the SAG1 with PD-L1 scfv and modified the location of HA tag within the construct. Finally, they verified the feasibility of recombinant T. gondii expressing anti-PD-L1 scFv by a series of in vitro validation experiments such as the binding assay, the neutralization assay and the mixture assay, indicating the engineered T. gondii as a promising strategy for cancer immunotherapy.
Recommendations for the authors
Major comments:
1. The author exploited multiple methods to verify the feasibility of engineered parasites with anti-PD-L1 scfv, including the binding assay, competition assay and etc. The author should validate the efficacy of the engineered agents by in vitro and in vivo co-culturing assay. For example, they should co-culture the engineered agents with T cells expressing PD-1 and tumor cells expressing PD-L1, and check whether their synthetic parasites work in the 2D co-culture system by measuring killing-related chemokines and cytokines (TNF-alpha, IL-6, IL-1beta, IFN-gamma etc.). Then they had better reproduce the same results in mouse model (B16F10/MDA-MB-231) by intro-tumoral injection.
2. In the Figure 4B and 4D, The author claimed that nearly 30% B16F10 cells treated with IFN-gamma could be GFP+ by binding with engineered parasites. However, the mixture assay results showed that nearly 90% B16F10 were GFP+. Why there is a huge variance? In addition, the author should control the variables and clarify the binding affinity of B16F10-mCherry cells (without IFN-gamma treatment) with engineered parasites as a control, not merely the non-fluorescent B16K1 without IFN-gamma treatment as a baseline.
Minor comments:
3. In the Figure 2, the author should supplement PD-L1 IF staining results as an orthogonal method.
4. In the Figure 2B, the author should clarify the statistical significance between RH and RH-PD-L1-N/C.
5. In the figure 4A, the author should calculate the MFI from each group and make a chart in the right side of flow cytometry data.
Comments on the Quality of English Language
NA
Reviewer 3 Report
Comments and Suggestions for Authors
The data demonstrated the possibility to use T. gondii to display functional scFvs on the surface and consequently deliver the parasite to the target cells and this has a moderate, but statistically significant, inhibitory effect…but what a difficult task was to read the manuscript. Apart from the many inaccuracies and grammar mistakes, it is the style to be heavy. The authors could, indeed should, summarize their Results in one third of the present length, making the content easier to digest, simpler to follow and by far more appealing. Therefore, I’d suggest to squeeze the text, check its language level carefully and only after these steps propose the manuscript again.
The authors should thoroughly revise the text because the language used in some sections is totally inadequate (and sometimes even the formatting must be revised). I did not report all the mistakes, but here are listed some examples from the first page:
Title: “Single Chain Fragment Variable”, it should be single chain variable fragment
Abstract: “surface-expression”, it should be surface-display (the expression is at the ribosome level), “constructions” should be constructs (the mistake has been repeated in other places in the text), “scFv showed better functionality…” maybe something like “the binding results suggested that the xxx scFv had more reliable functionality…”? line 19 is not understandable; lines 20-22 should be something like: “T. gondii expressing anti-PD-L1 scFv bound PD-L1 expressing cancer cells according to the level of available biomarker”; it should be “A mixed cell assay” (adjectivation of a noun does not take the plural) and then “…targeted 93% of the PD-L1 positive cells, with negligible…”; lines 25-27…compared….could inhibit…potentiated…(either all present or all past tense). Line 36: remove “however”; Line 38: “…it has been reported…”; Line 40: “…that produce…” (see above, present/past issue).
3.1: Please, explain more clearly the design of the constructs and why the differences among the constructs might be relevant.
Line 389: “To analyze if the anti-mDEC205 scFv or scFv-SAG1…”: are these the clones DC2 and DC2-SAG1? If yes, avoid misunderstanding by always using the same names.
Minor points:
Lines 103-105: the authors did not try expanding the repertoire of the binders, but of the antigens.
Lines 159-160: “Two expression cassettes were constructed to constitutively express proteins in T. gondii using previously described functional sequences: 5’ αTub promoter with 3’ SAG1 UTRs”. I cannot understand this sentence.
Lines 232-233: “1 μL of 2-fold serially diluted tachyzoite crude lysates (prepared in SDS-PAGE sample buffer) were applied” should be: “1 μl of 2-fold serially diluted tachyzoite crude lysates (prepared in SDS-PAGE sample buffer) was applied..”
Line 279: eliminate 2.8.1
Lines 316-317: “100 μL of medium containing 10% of 3-(4,5-Dimethyl-thiazol-2-yl)-2,5-Diphenyltetrazolium Bromide reagent (MTT, Invitrogen) was added…” should be: “100 μl of medium containing 10% of 3-(4,5-Dimethyl-thiazol-2-yl)-2,5-Diphenyltetrazolium Bromide reagent (MTT, Invitrogen) were added…”
5. Perspectives: the whole paragraph presents several grammar inaccuracies that should be removed.
Comments on the Quality of English LanguageThere are some mistakes and inaccuracies, but the major problem is the style.
Round 2
Reviewer 3 Report
Comments and Suggestions for Authors
no further comment